# Muted Voices: The Underrepresentation of Women in COVID-19 News in Portugal

**Rita Araújo** [1,*] , **Felisbela Lopes** [1], **Olga Magalhães** [2] **and Carla Cerqueira** [3]

1   Communication and Society Research Centre, University of Minho, 4710-057 Braga, Portugal; felisbela@ics.uminho.pt

2   Center for Health Technology and Services Research (CINTESIS), University of Porto, 4200-450 Porto, Portugal; olgamagalhaes@med.up.pt

3   Centre for Research in Applied Communication, Culture, and New Technologies (CICANT), Lusófona University, 4000-098 Porto, Portugal; carla.cerqueira@ulp.pt

*   Correspondence: ritaaraujo@ics.uminho.pt

**Abstract:** During the COVID-19 pandemic, the Portuguese media seemed to contribute to the symbolic annihilation of women. In spite of the fact that women play leading political roles as the Minister of Health and the Directorate-General of Health, women were almost mute in the COVID-19 news that was published in the Portuguese daily national press. In a sample of more than 6000 news sources, women account for less than 20% of them. Their lack of visibility in the news deepens the existing asymmetries of gender and amplifies the glass ceiling. The aim of this study was to analyze the media coverage of COVID-19 through a content analysis of the news that was published in two Portuguese daily newspapers with different editorial lines. Our period of analysis corresponds to the emergency-state periods (18 March to 2 May 2020; 9 November to 23 December 2020; 15 January to 28 February 2021).

**Keywords:** COVID-19; women; news; news sources; Portugal

## 1. Introduction

The first reports of what later became known as SARS-CoV-2 began in China in December 2019, and, a few months later, the World Health Organization declared a pandemic. Portugal was hit with the first case of SARS-CoV-2 infection on 2 March 2020, and, two weeks later, the government suspended all in-person activities from kindergarten to the university level, and the president declared an emergency state. The country lived in an emergency state from 18 March to 2 May 2020, which led to a severe lockdown. Later, in 2020, Portugal entered a new emergency state, from 9 November 2020 to 28 February 2021. Previous research indicates that the Portuguese media played a central role during the first lockdown in guiding citizens towards preventive behaviors, and that it became a means of fighting the pandemic (Araújo et al. 2021).

In order to understand the media coverage of COVID-19 in the Portuguese press, we conducted an analysis of the news that was published by two Portuguese daily newspapers with different editorial lines (*Jornal de Notícias* and *Público*). In order to ensure a broad analysis, we chose three time periods: from 18 March to 2 May 2020; from 9 November to 23 December 2020; and from 15 January to 28 February 2021. These dates coincide with periods during which the country had declared states of emergency and, therefore, they are comparable time frames. Our corpus of analysis is composed of almost 3000 news pieces, and 6370 news sources. However, female sources make up less than 20% of the total.

Women officials are underrepresented in the news about COVID-19, despite the fact that the leading national authorities are female. Moreover, the expert sources are also predominantly male. Our findings indicate an underrepresentation of women in the news content that is in line with other national and international studies that point out the near invisibility

of women in the news (see, for example, Global Media Monitoring Project—GMMP 2020). Since media agenda-setting is strongly influenced by news sources, and because the news media contribute to the representation of reality, which is in line with other studies (e.g., Fletcher et al. 2021; Jones 2020; Kassova 2020; Smith 2020), our research shows that there is a gender bias when it comes to COVID-19 news. This lack of visibility of women in the news contributes to a lack of recognition of them as professionals and experts, which deepens the asymmetries of gender.

Even though the news media coverage of the pandemic has promoted significant changes in the selection of sources, which has resulted in the higher visibility of expert sources, there has not been a true change in terms of gender representation. The choice of news themes has also contributed to the invisibility of women in the public media space, since the structuring of social fields, such as politics and economics, mostly belongs to male voices.

*1.1. Theoretical Framework*

1.1.1. The Role of News Sources in Setting the Agenda

The news media play a central role in the structuring of the public sphere, as they are mediators between the public and institutions. This watchdog function of journalism is undertaken on behalf of the citizenry, and the journalist is perceived as "a mediator between the citizen and the politician, the former's representative before power, who ensures that the voice of the public is heard" (McNair 2009). Therefore, journalism is closely linked to power, "and the way that journalists deliver news information has a profound impact on the shaping of the public and private debate" (Mellado 2015, p. 597) with regard to diverse themes. Ericson et al. (1989) argue that "news is a representation of authority", which means that journalists give a voice to those who are legitimized to speak—for themselves or on behalf of a group. Indeed, news is a product of the transactions between journalists and their sources (Ericson et al. 1989), and while, theoretically, news sources can come from anywhere, in practice, "their access to journalists reflect the hierarchies of nation and society" (Gans 2004). This happens because, alongside of the resources and skills that are required to be "newsworthy", many of these "go hand in hand with economic power, and are possessed by only a few" (Gans 2004).

Nonetheless, journalism should prime for the plurality and diversity of news sources in order to convey different realities and points of view on a given subject. "The diversity of sources in the news reflects the extent to which the media are able to function as an arena for participation in public debate" (Sjovaag and Pederson 2019, p. 215), and "it has been argued that journalistic quality is affected when the media do not meet their responsibility to include a variety of sources and viewpoints to a pluralistic society" (Rodgers and Thorson 2003, p. 672). Indeed, despite the fact that journalism should promote pluralism and the diversity of voices and social groups, some studies show that the "selection of news sources is still biased" (Swert and Hooghe 2010, p. 70), and that women are underrepresented in the news (Armstrong 2006; Byerly 2016; GMMP 2020; Len-Ríos et al. 2005; Ross 2007, 2009; Van Zoonen 1988). In the introduction to *Gendered News*, a Special Issue of *Journalism*, Cynthia Carter states that "journalism plays a central role in shaping our perceptions of gender relations, sometimes conferring—and sometimes denying—public recognition to people purely on the basis of sexual difference" (Carter 2005, p. 259).

The relationship between women and the media, or between gender and the media, has been a topic of continuing discussion in journalism (Carter et al. 1998; Chambers et al. 2004; De Bruin 2000; De Bruin and Ross 2004; Gill 2007; Steiner 2017). The access of news sources to the media provides them with the opportunity "to gain notoriety", and this is especially true when it comes to political actors who can thereby set the political agenda, and "so the lack of women on the news programs carries several implications for their representation in the political arena" (Baitinger 2015, p. 580). In the area of health, for instance, "it has been argued that journalists rely more heavily on sources and experts because of a focus on novel health findings and the technical nature of the information"

(Len-Ríos et al. 2009). Moreover, although "journalism has been under pressure to make use of more specialists" (Gans 2004, p. 143), the truth is that specialist sources are not all equally important to journalists. A study that analyzed US local-television-news pieces found that male expert sources are cited significantly more often than female expert sources in the news stories (Desmond and Danilewicz 2010), and data from GMMP (the Global Media Monitoring Project), which is the world's largest global media monitoring project, show the same phenomenon from 1995 to 2020 (GMMP 2020). There have obviously been advances; however, they are not significant when we look at the roles that women occupy in the most diverse spheres. The gender imbalance in the news media is visible, but it is worse when we are talking about authorities and experts.

These data are especially relevant when we consider that news sources have an important role to play in setting the media agenda. They "feed" news stories to the media, and they help them to make sense of technical and specialized information.

The relationship between reporters and news sources has been studied within news sociology, and it has often been depicted as a struggle for power (Sigal 1973; Mencher 1991). Leon Sigal (1973) was a forerunner in this field and, in the 1970s, he conducted an analysis of the news that had been published in the press over 20 years. Sigal claimed that the news content in the press depends on the proactivity of the sources, and on what they convey through informal, routine, or initiative channels. Years later, Melvin Mencher (1991) wrote that "the source is the reporter's lifeblood". "Without access to information through the source, the reporter cannot function" (Mencher 1991, p. 283). News sources add credibility to the news story, and journalists usually learn about events through news sources. Since journalists report on facts and should refrain from giving their personal opinions, they turn to sources to provide them with arguments for or against a given topic.

1.1.2. Women (Barely) in the News

Even though "women's minority presence in the news is widely established", "its roots are not yet known" (Baitinger 2015, p. 580). Nonetheless, several authors have pointed out reasons that may be behind the underrepresentation of women in the news. A study of gender representations in Norwegian newspapers states that "the reason for gender imbalances in the news partly lies with journalism itself" (Sjovaag and Pederson 2019, p. 216). The same researchers explain that journalism is attracted to power because in power lies information, and the people in power are mainly men. Indeed, it is true that "the news is indelibly linked to gender and power" (Len-Ríos et al. 2005, p. 154). The "glass ceiling" effect is a known metaphor that describes the difficulties that women face when trying to achieve higher hierarchical positions, and even though it is often related to the political field or to leadership positions, it can be found in several social domains. When it comes to politics, gaining access to the news media provides political actors with "the opportunity to gain notoriety on important policy issues", which means that the underrepresentation of women in the news leads to their underrepresentation in the political arena itself (Baitinger 2015, p. 580). The truth is that "news content does not accurately reflect politics and its (female) actors" (Vos 2013, p. 390). Women politicians, and particularly leaders, commonly experience far more gendered and personalized critical media coverage (Williams 2017).

Van Zoonen states that the dominant view on women and the news is that "the underrepresentation of women in news production leads to the underrepresentation of women in news content". This premise implies that "news content will change with an increase in the number of female journalists" (Van Zoonen 1988, p. 37). However, and "even though the number of female journalists has been consistently increasing (the so-called "feminization" of the profession), this has not improved women's equal participation in media or gender balance within media content" (Lobo et al. 2017, p. 1149). Indeed, over the past decades, there has been an increase in the number of women who are studying journalism and working as journalists (de Vuyst and Raeymaeckers 2019), even if "women journalists are still significantly underrepresented in older age groups, in decision-making

positions and in prestigious newsbeats and media sectors" (de Vuyst and Raeymaeckers 2019, p. 24). Portuguese newsrooms seem to be no exception, as they present low numbers of women in decision making in news outlets (Subtil and Silveirinha 2017; Lobo et al. 2017). Furthermore, as applied to the news profession, "research examining male and female reporting differences has been inconclusive" (Rodgers and Thorson 2003, p. 659). Even though some studies show that female reporters use more female sources, other studies have found no differences between male and female reporters when it comes to sourcing. "The mix of findings leaves questions about whether male and female reporters differ on important reporting aspects or whether increasing the number of women reporters will change news content in noticeable ways" (Rodgers and Thorson 2003, p. 659).

Len-Ríos et al. (2005) claim that "one explanation for the underrepresentation of women is that reporters preserve hegemonic cultural norms" (p. 152). Therefore, "when journalists cover stories deemed newsworthy by political and societal elites, most of whom are males, journalists reproduce societal norms privileging men" (Len-Ríos et al. 2005, p. 152). This happens because the public media space is an arena where "(hegemonic) meanings of gender have continuously been negotiated, reinforced, and/or challenged" (Santos et al. 2018, p. 2). Hence, the news media reproduce dominant versions of social reality, which consequently reinforce structural inequalities (amongst which are gender inequalities). Therefore, news decisions are embedded in the tensions, divergences, and contradictions that take place in media industries, such as the personal idiosyncrasies of the professionals, (male-biased) newsroom culture, organizational and economic constraints, media ownership, media regulation, etc. (Carter and Steiner 2004; North 2009; Santos et al. 2018).

## 2. Materials and Methods

The goal of this research was to analyze the media coverage of COVID-19 in the Portuguese press, and especially the themes of the news and the news sources that journalists resort to the most. We analyzed all COVID-19 news published in two Portuguese daily newspapers that follow different editorial lines (*Jornal de Notícias*, a popular newspaper, and *Público*, a quality newspaper). The editorial orientations promote differences in the media coverage and, namely, in the selection of and the approach to themes and news sources. Even though journalism, in general, should promote the diversity of news sources, this is a characteristic that quality media, in particular, should pursue. *Jornal de Notícias* is a popular newspaper that was founded in 1888 and that is owned by the Global Media Group. It is the only national daily newspaper with a main newsroom that is located in Oporto, which is the second largest Portuguese city. *Público* is a quality newspaper that was founded in 1990, and it is owned by a large economic group named Sonae.

Data was collected from the digital (paid) versions of the chosen newspapers, and this choice was made by taking into account the data from the Portuguese Association for the Control of Circulation (Associação Portuguesa de Controlo de Tiragem) that pertain to the digital paid circulation in the last four months of 2020. To ensure a broad analysis, we chose three time periods: from 18 March to 2 May 2020; from 9 November to 23 December 2020; and from 15 January to 28 February 2021. These dates coincide with periods in which the country declared states of emergency, and, therefore, they are comparable time frames. Case selection included all news texts published in the sections "Primeiro Plano" ("Forefront Section", *Jornal de Notícias*) and "Destaque COVID-19" ("COVID-19 Highlights", *Público*). These sections were chosen because they allow for comparisons since they consist of all news that is deemed the most relevant for these newspapers. Collected data were then coded and categorized through SPSS, and content analysis followed an analysis grid that was previously elaborated on and tested by researchers. With regard to the news texts, researchers wanted to know what the most common news genre was, the place of the news, the positioning of the title, and the most common news themes. As for the news sources, researchers wanted to understand the sources that journalists most talked to, their statuses and their affiliated institutions, the place of the news source, and their medical specialties, when applicable. For the purpose of this study, we will focus on news texts.

To fulfill the aim and the purpose of the study, a set of research questions (RQ) were raised, namely:

RQ1: "The choice and use of news sources is influenced by the newspaper's editorial line";

RQ2: "The presence of two women in leading political decision roles (in the Ministry of Health and the National Public Health Authority) promoted an effect of contagion towards the choice of women as news sources";

RQ3: "The visibility of expert sources in the news media increased women's presence in the news".

## 3. Results

Our corpus of analysis is composed of 2929 news pieces: a total of 1850 were published during the first emergency-state period; 457 during the second; and 622 during the last. News is consistently the predominant genre, and it accounts for 74.9% of the texts during all of the periods of analysis. The analyzed texts quote 6370 news sources, and the journalists talked to an average of two sources per news piece. Female sources represent 18.1% of the total, whereas document sources account for 12.8%, and male sources are predominant (50.3%). When it comes to the geography of news sources, they are usually national representatives (52.6%). International news sources account for 16.2%, and, among those, European sources are the most common (8.9%).

During the COVID-19 pandemic, both the Portuguese Minister of Health and the Portuguese Director-General of Health were women, and they attracted a great deal of media attention. During this period, the news media valued expert sources, and particularly scientists and doctors, which are groups that, in Portugal, consist of a significant number of women (OECD 2019). Nonetheless, women were undervalued in the SARS-CoV-2 news pieces. During a time where the consumption of news increased greatly, the news media lowered the glass ceiling, through which women are not allowed to become actors who contribute to the (re)configuration of the public sphere. Moreover, if we look into the most structuring themes of the public space—political decisions—the gap between men and women is even more pronounced (56.4% of the news sources in national politics are men, compared to 14.3% who are women). During that time, women became visible as nonprofessional sources or common citizens, and journalists expected them to provide mere testimonies in a situation that was already being led by other social agents. This result refocuses the discussion onto an issue that has long arisen in the case of gender studies, which is the association of men with the domain of public space, and women with the domain of private space (McLaughlin 1999).

Looking at all the analyzed news, female sources account for less than 20% of the total (18.1%), even though the latest data for the Portuguese population indicate that women represent 53% of the inhabitants. These numbers show an underrepresentation of women in the news about COVID-19 since they make up the majority of the Portuguese population and, still, they do not seem to be present in the public media space. The latest data by the GMMP (2020) had already shown these asymmetries, with men representing 89% of the sources in the written press, radio, and television when the news topics were celebrities, arts and media, and sports. In the news about politics and government, male voices represent 73%; in science and health, they represent 68%; in economics, they represent 60%; and in crime and violence, they represent 59%. Women were only more visible than men in the news when the topics were social or legal topics (52%), which deal with development issues, social and legal issues, and human rights. When compared to the data from 2010, the representation of women in this topic increased by 20%. With regard to the science and health news, there was a decrease of 7% (from 29% in 2010, to 32% in 2020). In the politics and government news, there was a slight increase throughout these 10 years: from 25% in 2010, to 27% in 2020.

### 3.1. Public Men and Private Women

When we look into human news sources, which are composed of five macrocategories (official sources; expert sources on behalf of groups; expert sources on an individual level; nonexpert sources/citizens; and others), our analysis shows that women are more represented within the nonexpert sources/citizens category (44% women compared to 51.5% men) (Table 1). This contributes to a well-known dichotomy in the news: the gap between public men and private women. Men are asked for decisions, thoughts, and strategic action; women are required to give testimonies, to expose feelings, or to describe a physical state. Even though this is the year 2020, Victoria Camps' intent is still unfulfilled (Camps 1998). The author stated that the 21st century would belong to women; however, the Portuguese public media space does not reflect this, and neither do other geographies. Several international studies, in diverse periods and contexts, show that media opinions are dominated by men's voices and perspectives (Silveirinha 2004; Perry 2009; Gallego 2005; GMMP 2020), and this does not seem to be a changing reality. Thus, this traditional mode of distinguishing private from public within the media context legitimizes women's oppression (Magalhães et al. 2012).

**Table 1.** Human news sources quoted in COVID-19 news pieces (absolute numbers).

| News Sources' Status | Women | Men | Group | Nonidentified | Total |
|---|---|---|---|---|---|
| Official | 413 | 1179 | 39 | 110 | 1741 |
| Experts—on behalf of a group | 274 | 1078 | 22 | 39 | 1413 |
| Experts—individual | 202 | 456 | 16 | 24 | 698 |
| Nonexperts/Citizens | 212 | 246 | 20 | 0 | 478 |
| Others | 53 | 219 | 12 | 59 | 343 |

While some studies recognize that the underrepresentation of women is related to their professional roles, which means that women in power positions are less common than men (Len-Ríos et al. 2005), our corpus also shows an underrepresentation of women in specialized and professional roles. Indeed, female voices represent 22.5% of expert sources, compared to 72.7% of male sources in the same category (Figure 1). On the one hand, women are more visible among the expert sources who talk on an individual level (28.9% compared to 65.3% men), and they are more represented in the following fields: technicians, nurses, university professors within natural sciences, and psychologists/social assistants. On the other hand, women are pushed into a spiral of silence in the groups that are composed of university professors within health, as well as doctors. These results are in line with those from the GMMP (2020), which have shown a decrease in representations of women in these areas in the past 10 years.

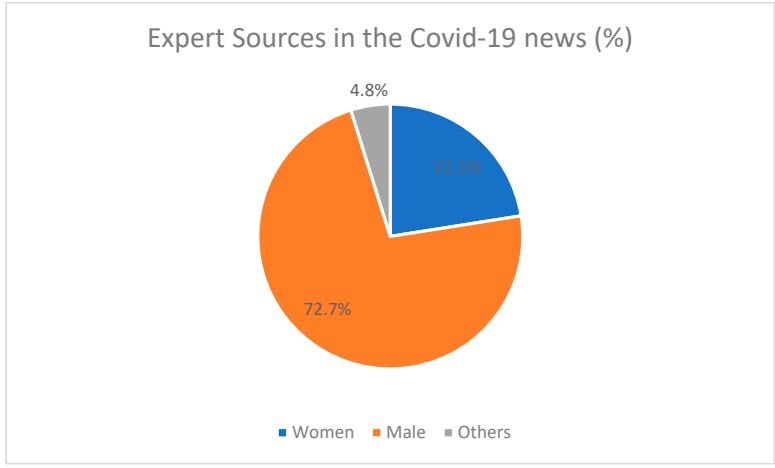

**Figure 1.** Percentages of expert sources in the analyzed COVID-19 news by gender.

When we look into expert sources who talk on behalf of groups, women's voices represent even less (19.4%), whereas men account for 76.3% in the same category. A closer look tells us that women are more visible than men when they are the heads of university departments or research centers, or when they are nurses, pharmacists, or press officers; these categories represent exceptions.

According to 2018 data from Pordata, Portugal had 35.283 university professors: 19.368 who were men (54.9%), and 15.915 who were women (45.1%). In the analyzed COVID-19 news pieces, journalists showed a preference for expert sources and, among these, doctors are the most visible (15.4%). Within this group, women represent only 21.5% of the total. Data from 2021 show that women are predominant in health professions in Portugal, and that they also make up the majority of the Portuguese population (CIG 2021). The second group of specialized sources that the journalists talked to the most is university professors (14.6%). Again, women sources account for 20.3% of the sources in this category. Even though the latest available data indicate that women represent 56% of the doctors in Portugal, our research shows that female doctors are not very visible in the media. These results are consistent with our previous research, which, besides showing the high visibility of doctors in the news compared to other health professionals, also points out the underrepresentation of women in the news media (Araújo and Lopes 2017; Araújo 2016). Our findings are also in line with several international studies that highlight the underrepresentation of women in the news (Len-Ríos et al. 2005; Van Zoonen 1988) and the higher visibility of male expert sources compared to female expert sources (Desmond and Danilewicz 2010). Media agenda-setting is highly influenced by news sources and, since the news media contribute to a representation of reality, these findings indicate that there is a gender bias within the analyzed health news. Moreover, this study shows us that "women appear for what they are and men for what they do" (Gallego 2013), and that women continue to be absent as the protagonists of certain fields (in this case, as specialists in the field of health). These results are in line with other international studies that demonstrate that, as voices become more prominent and well known, the female share in them decreases (Jones 2020).

As for official news sources, women represent 23.7%, and this number is due to the regular media presence of both the Portuguese Minister of Health and the Director-General of Health, who represent 1.5 and 1.7% of the news sources, respectively. These two women were probably the leading figures of the COVID-19 pandemic in Portugal, as they were present at the majority of press conferences and official appeals to the nation. However, their presence in the public space is not matched by their presence in the news media. The Minister of Health, Marta Temido, was quoted 95 times, according to our analysis, and the Director-General of Health, Graça Freitas, was quoted 106 times. Within official sources, the differences between women's and men's representations are colossal: female news sources represent 23.7% of official sources, whereas men account for 67.7%. This result is very interesting because it shows us that, even when they are in leadership positions, women do not have an active voice in the news media, and this was even more so during the pandemic, when many of the protagonists in the political sphere and in the area of health at the (inter)national level were women.

### 3.2. Political Men and Invisible Women

When it comes to journalists' choices of news sources, they often choose those who are more socially valuable, and here is where women become more invisible. The same happens when we cross the gender of news sources with news themes. The most structuring themes become news through male voices. Men design political decisions, they organized health services during the pandemic, and they control the economy. This is the current state of events, and the news media lowered the glass ceiling even further, which prevents women from achieving high positions.

When we analyzed the gender of the news sources in relation to the news themes, our data show that women are underrepresented in all social fields (Table 2). When looking at

politics, for instance, the gender gap is colossal: only 14.3% of the news sources that were quoted by journalists on national politics are women, compared to 56.4% of whom are men.

**Table 2.** Gender of news sources in relation to news themes (%).

| News Themes | Gender of News Sources | |
|---|---|---|
| | Women | Men |
| Prevention | 20.9% | 48% |
| Medical acts | 21.2% | 50.6% |
| Research and Development | 18.6% | 43.8% |
| Organization of services | 12.1% | 46.7% |
| Situation Portrayals | 21.7% | 47.5% |
| National Politics | 14.3% | 56.4% |
| International Politics | 11.7% | 52.4% |
| Economy | 14.7% | 48.4% |
| Society | 19.6% | 52.7% |
| Professional Situations | 25.8% | 44.8% |

When the theme is international politics, female sources account for 11.7%, whereas male sources represent 52.4%. Indeed, this is the news theme where women are less visible, and it is one of the categories where the gap between men and women is wider. Political decisions are one of the most structuring themes of public space, and our analysis shows that women do not have an active voice in it—despite the fact that the main political actors during this pandemic were women. Indeed, not only are the Portuguese Minister of Health and the Portuguese Director-General of Health women, but so is the President of the European Commission, Ursula von der Leyen, and the European Commissioner for Health, Stella Kyriakides. We already saw that access to the news provides political actors with the chance to gain notoriety, and, therefore, women seem to be at a disadvantage, since our analysis shows that they are clearly underrepresented in politics-related news.

Even in situations where women are expected to give testimonies, such as situation portrayals or society-related news themes, female voices are undervalued, and they represent less than half of male voices (21.7% in situation portrayals, and 19.6% in society-related themes). This is also an interesting aspect that shows that women's perspectives are often not taken into account in the news.

The same silence happens with news themes that are closer to the health or science fields, which is the case for medical acts (where female voices account for 21.2% of the news sources) and research-and-development news (18.6% of women)—in both of these categories, men represent half, or almost half, of the news sources. Nonetheless, women are more visible in the news pieces on medical acts, social portraits, and professional situations—with the latter becoming news when there are problems that affect nurses and/or doctors. Women professionals are portrayed as victims of circumstances, instead of as decisionmakers. This result reminds us of a phrase by Parry-Giles (2000), which is referred to by Rosalind Gill (Gill 2007, p. 120), who says that "we should fear women with power, but admire women with victim status".

## 4. Discussion

During the COVID-19 pandemic, the Portuguese news media seemed to contribute to the symbolic annihilation of women (Tuchman 1978). The term "symbolic annihilation" refers to the erasure of women in the news, despite the fact that they make up the majority of the population in Portugal, and they are in positions of power at the political level or are even professionals in health and science. It is important to stress that we refer to symbolic annihilation because this is what happens when a group's lack of representation affects its real-life power in the public sphere. Women's lack of visibility in the news also generates their lack of recognition as professionals, and as voices who are able to discuss the central issues in the public domain. This is one of the asymmetries that has been highlighted many

times in the fields of gender and media studies, but which seems to persist. Indeed, the glass ceiling, which does not allow women to achieve power positions, is present in several social fields, and it is amplified by journalism, the same journalism that should represent society in a balanced, plural, and neutral way, and that has an important role to play in the achievement of gender equality and diversity in societies. Our data show that journalists amplify women's invisibility, to quote Pionchon and Derville's (2004) impressions on the reduced presence of women in politics.

The Portuguese Minister of Health and the Director-General of Health played central roles in the management of this pandemic, and they became sort of "alibi women" (i.e., women who legitimize the gender imbalance that is well represented in our analysis). Furthermore, their public visibility, which is due to the positions that they hold, may also contribute to the perceptions that they are present in central areas, and that gender inequalities no longer exist.

Even though the news media coverage of the pandemic has promoted significant changes in the selection of sources, which has resulted in the higher visibility of expert sources, there has not been a true change in terms of gender representation. On the contrary, the visibility of women is still really low, despite their presence in central political roles as the Minister of Health and the Directorate-General of Health. The choice of news themes has also contributed to the invisibility of women in the public media space, since the structuring of social fields, such as politics and economics, mostly belongs to male voices. Thus, these results continue to show that "news does not reflect reality but presents a consistently more male-dominated view of society than exists in reality" (Gill 2007, p. 115).

When we take a close look at the news media coverage of the pandemic throughout 2020 and 2021, our data indicate that men had significant control over (political) decisions. Symbolically, this is visible in the Portuguese government's actions: when it comes to the daily management of the disease, the presence of the Minister of Health and the Director-General of Health was notorious in the media; however, when the time came to communicate difficult decisions that implied structural changes to society, such as lockdowns or restrictive measures, the prime minister would monopolize the media space. This became a symbol of what was already a social and media trend of men making political decisions and women running those decisions: men thinking, and women feeling/performing/taking orders.

The news media are crucial in structuring the public sphere, since they work as mediators between the public and social institutions. When choosing news sources, journalists should take into account the plurality and diversity of voices that can contribute to the portrayals of different social realities and their complexities. The access of news sources to the news media and, therefore, to their visibility in the news, contributes to their public notoriety. In the media coverage of political themes, this notoriety is even more relevant. This means that news sources that do not appear in the news do not become politically relevant, and, therefore, the media may contribute to the glass ceiling effect that prevents women from achieving high and powerful positions within society. The role of journalism is also to deconstruct social inequalities, such as gender inequality, and therein inheres its role in social responsibility (Vega Montiel 2014). Our analysis seems to indicate that the Portuguese news media did just that by shedding light on male voices and pushing female sources out of the news—even when they were high representatives of the country during the COVID-19 pandemic, such as the Minister of Health and the Director-General of Health.

## 5. Final Remarks

The pandemic promoted some changes in journalism, and, namely, in news-production processes and in the themes and sources in the news (Lopes et al. 2021). These changes were a result of several lockdowns and restrictions, and they moved expert sources from the silent margins into the public media discourse. However, the Portuguese news media continued to promote a gender bias in which male sources were more valued than female ones. This is

especially relevant since, on the one hand, two of the main decisionmakers were female and, on the other hand, the most quoted groups of sources (doctors and researchers) are mostly composed of women. Women represent 56% of doctors and 45.1% of university professors. Following social responsibility, journalism should be concerned with the promotion of gender-balanced coverage. According to our hypotheses, we conclude that:

RQ1: The editorial lines of the newspapers did not influence the choice and use of news sources. Our analysis shows the same pattern of news-source choices in both media outlets;

RQ2: The presence of two women in leading political decision-making roles, such as in the Ministry of Health and the National Public Health Authority, did not promote an effect of contagion towards the choice of women as news sources during the pandemic. What is more, these two women, themselves, did not have a significant media presence;

RQ3: The presence of women in the expert categories of news sources did not reflect an increase in their presence in the news.

These data show that journalism is still reproducing gender imbalance, even though 48.2% of the journalistic profession is composed of female journalists (dos Jornalistas 2022). Such an imbalance can be found before and after the news-production process. The (small) presence of women in the news may be explained by their equally small presence in power roles: usually, women are not the heads of services within hospitals, nor are they at the top of academic careers, compared to their male peers. In Portugal, women's quotas are mandatory in politics, in public companies, and in companies that are listed on the stock exchange in order to promote gender equality[1]. However, quotas, just as other affirmative-action measures, are marked by advances, resistance, and misunderstandings (Cerqueira et al. 2021). Moreover, the gender imbalance is still a reality in several social fields (CIG 2021). Still, journalism conveys a small group of elite sources whose voices are reproduced through their skills in constant self-reproduction (Lopes 2011).

Indeed, the news arena is still a kind of "no women's land", in which the way in is highly selective. Women do not have the same opportunities as men to become news sources. As such, the few women who are represented in the public media space have to deal with male dominance and with the patriarchal gender ideology that still rules in Portugal. Both the news media and media researchers may contribute to the reinforcement of some of these patriarchal actions by not questioning them. As the article by Ross et al. (2020) shows, there are no easy solutions to improving the visibility of women politicians in the media landscape, and more studies are needed to unravel this reality and to enable changes in the medium and long terms. More than 25 years after the Platform for Action that resulted from the Beijing Conference and that placed the need to look at the relevance of the media in the deconstruction of gender inequalities, it is still urgent to look at the resistances and the contradictions that exist in news coverage, and that continue to reproduce gender inequalities. It is in this sense that we underline that the data that result from this study show that the news media continue to operate a symbolic annihilation of women. "Only when women are visible in the media as playing active roles in politics, at work and in the affairs of ordinary life, can they also constitute themselves more actively as publics to be encouraged to participate in all areas of social life in a recognizably equal way" (Silveirinha 2009, p. 7).

**Author Contributions:** Conceptualization, R.A. and F.L..; methodology, R.A. and F.L.; formal analysis, R.A.; investigation, R.A., F.L., O.M., C.C.; resources, R.A., F.L, O.M.; writing—original draft preparation, R.A., F.L., C.C.; writing—review and editing, R.A and C.C.; visualization, R.A. and C.C.; supervision, R.A.; project administration, R.A.; funding acquisition, R.A. and F.L. All authors have read and agreed to the published version of the manuscript.

**Funding:** This work is financed by national funds through FCT—Fundação para a Ciência e a Tecnologia, I.P., under the project UIDB/00736/2020 (base funding) and UIDP/00736/2020 (programmatic funding), and within the scope of the contract signed under the transitional provision laid down in article 23 of Decree-Law no. 57/2016, of August 29, as amended by Law no. 57/2017, of July 19.

**Institutional Review Board Statement:** Not applicable.

**Data Availability Statement:** Not applicable.

**Conflicts of Interest:** The authors declare no conflict of interest.

## Note

[1]  On 23 June 2017, the Portuguese Parliament approved a proposal by the government that forced public companies and companies that were listed on the stock exchange to hire more women on supervision and administration boards. In public companies, at least 33.3% of these positions need to be fulfilled by women; in listed companies, this percentage decreases to 20%.

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
