# Peer review of "Muted Voices: The Underrepresentation of Women in COVID-19 News in Portugal"

_socsci, doi:10.3390/socsci11050210_

Round 1
Reviewer 1 Report
The article is well composed and written. It addresses an important and relevant topic and provides a valuable insight into general women underrepresentation in media.
I do however have some issues with how author(s) structure their argument.
The article is strong on descriptive side but I find it somewhat lacking in the ‘explanatory department’. The is no ‘Literature review’ section of the article. I assume that lines 50 and 113 are supposed to be the separate sections of the article technically constituting a review part. Yet we find no hypotheses by the author(s) at the end of this section. There are theories cited and some provisional explanations of women underrepresentation given but at the end we do not learn which one of those are endorsed or tested by the author(s).
The article could greatly benefit by careful explanation of the background against which underrepresentation of women is measured. I do not think that measuring versus general sex proportion in population provides any fruitful insight into the matter at hand (212-213). If it does one could argue that such an underrepresentation may be explained by the fact that circa 60% of COVID death cases are male.
Further in the ‘Result’ section we get some glimpse at other possible backgrounds for the measurement when we learn that 56% of doctors in Portugal are female. I wish authors had gone that way further. One can asses the level of underrepresentation of women among experts only after assessing the male/female proportion in relevant group of possible experts (i.e. doctors with academic credentials and research practice). I strongly advise deep restructuring of the argument in that respect.
I am also disappointed by the lack of more media oriented analysis. There are two newspapers chosen for the study. Their differences are only barely touched upon (167-168) and there is no analysis of the differences of the levels of underrepresentation between them.
I would also advise some moderation in language department. ‘Symbolic annihilation’ (4, 314), ‘invisibility’ (39) seems to be a rather excessive terms in the context of 20% representation. Moreover ‘annihilation’ suggests that something pre-existing is being destroyed, and yet despite obvious drawbacks and unsatisfactory levels the media representation of women is improving.
Author Response
- Dear Reviewer,
- The Literature Review section of the article was not properly indicated, and we revised that. It goes from lines 51 to 165, and it is composed of two main themes (The role of news sources in setting the agenda, and Women (barely) in the news). We revised the underrepresentation of women and included more recent references.
- We followed the reviewer’s suggestion and included formal hypotheses that guided our work. Also, we underlined the differences between the two analyzed newspapers, which were explored in a hypothesis.
Reviewer 2 Report
This article about “Muted Voices: the underrepresentation of women in Covid-19 news in Portugal” is attractive. However, the authors need to change or improve the next questions:
-Theoretical framework. There are textual citations without number page. For example,
Years later, Melvin Mencher (1991) wrote that “the source is the reporter’s lifeblood”. “Without access to information through the source, the reporter cannot function” (Mencher, 1991).
Furthermore, a lot of citations are ancient. It is necessary to include more updated citations.
-Methods. It combines quantitative and qualitative tools. The authors analyze Covid-19 news published in two Portuguese daily newspapers which follow different editorial lines. They say they use SPSS, but the data are only descriptive, without inferential efforts.
-Results: in general, they are limited and descriptive. Indeed, it is necessary to introduce some charts or Figures to illustrate the results.
-Conclusions: There is a discussion, but not a conclusion. You have to include more limitations and future research lines.
Author Response
Dear Reviewer,
-We revised the whole text and provided number pages to all textual citations.
- We included more updated citations in-text
-We introduced some tables and figures in order to illustrate results and make the article more understandable.
-We reviewed the paper and included final remarks, taking into account the hypotheses.
Round 2
Reviewer 2 Report
The paper has improved and it can be published. The authors have to review formal English style.